# Organophosphorus Flame Retardant TPP-Induced Human Corneal Epithelial Cell Apoptosis through Caspase-Dependent Mitochondrial Pathway

**DOI:** 10.3390/ijms25084155

**Published:** 2024-04-09

**Authors:** Chen Chen, Daolei Cui, Jingya Li, Caixia Ren, Danlei Yang, Ping Xiang, Jianxiang Liu

**Affiliations:** Yunnan Province Innovative Research Team of Environmental Pollution, Food Safety and Human Health, Institute of Environmental Remediation and Human Health, School of Ecology and Environment, Southwest Forestry University, Kunming 650224, China; chenchen3462@163.com (C.C.); daolei_cui@126.com (D.C.); jingyali1999@126.com (J.L.); rencx107@163.com (C.R.); danleiyang1997@126.com (D.Y.)

**Keywords:** TPP, human corneal epithelial cells, cell apoptosis, mitochondrial apoptosis pathway, gene expression

## Abstract

A widely used organophosphate flame retardant (OPFR), triphenyl phosphate (TPP), is frequently detected in various environmental media and humans. However, there is little known on the human corneal epithelium of health risk when exposed to TPP. In this study, human normal corneal epithelial cells (HCECs) were used to investigate the cell viability, morphology, apoptosis, and mitochondrial membrane potential after they were exposed to TPP, as well as their underlying molecular mechanisms. We found that TPP decreased cell viability in a concentration-dependent manner, with a half maximal inhibitory concentration (IC50) of 220 μM. Furthermore, TPP significantly induced HCEC apoptosis, decreased mitochondrial membrane potential in a dose-dependent manner, and changed the mRNA levels of the apoptosis biomarker genes (*Cyt c*, *Caspase-9*, *Caspase-3*, *Bcl-2*, and *Bax*). The results showed that TPP induced cytotoxicity in HCECs, eventually leading to apoptosis and changes in mitochondrial membrane potential. In addition, the caspase-dependent mitochondrial pathways may be involved in TPP-induced HCEC apoptosis. This study provides a reference for the human corneal toxicity of TPP, indicating that the risks of OPFR to human health cannot be ignored.

## 1. Introduction

Organophosphorus flame retardants (OPFRs) are widely used as alternatives to brominated flame retardants (BFRs) in a variety of commercial products, including plastics, baby products, textiles, paints, and furniture [1,2,3]. OPFRs are mainly added to the materials physically; therefore, they can be easily released into the environment via volatilization, leaching, and abrasion [4]. Triphenyl phosphate (TPP) is one of the most widely used OPFRs and has been frequently detected in diverse environmental samples including soil, sources of water, and air [5,6,7]. In addition, TPP was detected in 69 indoor dust samples taken from floors and furniture in homes, offices, and daycare centers in urban Beijing, with an average concentration of 2004 ng·g^−1^ [8]. The presence of TPP was detected in air samples from eight locations in the urban, rural, suburban, industrial, and agricultural areas of Bursa, Turkey, with a concentration of 54~320 pg·m^−3^ [9]. The presence of TPP has even been detected in air collected in the Arctic, at concentrations of 31.3~239 pg·m^−3^ [10]. According to statistics, TPP is one of the most abundant OPFRs in air and dust samples worldwide [11]. This shows that we often come into contact with TPP in our daily life, but what is more noteworthy is that existing studies have found that TPP also exists in the human body. Schindler et al. (2013) detected the presence of TPP in the urine of passenger aircraft pilots and crew [12]. Kim et al. (2014) detected organophosphorus flame retardants in 89 breast milk samples from Japan, the Philippines, and Vietnam, and found that TPP was detected in more than 60% of the samples, and the lipid weight of TPP in the samples was up to 140 ng/g [13]. Guo et al. (2023) also detected the presence of TPP in human blood in some parts of China, and the content was relatively high, with a concentration of 0.346 ng/mL [14]. This indicates that TPP can accumulate in the human body, which is a potential threat to human health. Hence, the toxicity of TPP has attracted worldwide attention. However, the current reference dose for TPP health risk assessment (70 μg/kg) is still based on the threshold dose derived by Sutton et al. in 1960, Eastman Kodak Industrial Medicine Laboratory data [15]. In 2020, the US Environmental Protection Agency (EPA) listed TPP as one of the chemicals that requires a health risk assessment (Final Scope of the Risk Evaluation for Triphenyl Phosphate [S]. 2020).

Recently, most studies have mainly focused on how TPP induces neurotoxicity, reproductive and developmental toxicity, hepatotoxicity, etc. [16,17,18,19]. For example, Liu et al. (2020) suggest that TPP exposure could cause a degree of neurotoxicity by influencing redox balance, activate neuroinflammation mediated by microglia, and induce central nervous system cell apoptosis [20]. Wang et al. (2020) analyzed the toxic mechanism of TPP on human normal hepatocytes (L02) through multiple omics methods such as transcription, protein, and metabolism [21]. A study by Jannuzzi et al. (2022) found that TPP inhibited proteasome activity by inducing apoptosis of skin cells (HaCaT) and production of reactive oxygen species, thus causing damage to human skin [22]. Wang et al. (2021) confirmed that PPARγ plays an important role in the TPP-mediated lipid metabolism disorder through lipemic analysis in human choriocarcinoma-derived placental JEG-3 cells [23]. However, TPP can generate various forms of incidental contact with the human body from different microenvironments, thus causing it to come into unavoidable contact with the eyes, which may have detrimental effects.

The eye is an important visual organ of the human body. The ocular surface including the cornea and conjunctiva and its overlying tear film is the outermost barrier that protects the eyeball from environmental damage, while the human corneal epithelium is the outermost cell on the eye surface, and continuous daily exposure to contaminants can cause corneal damage and corneal inflammation [24,25]. Epidemiological data show that the number of outpatient visits for ocular surface diseases, such as allergic conjunctivitis and corneal inflammation, is correlated with air pollution; this is because the tear film on the cornea attracts particles from the air, causing pollutants attached to the tear film to come into contact with the cornea, inducing damage [26]. Our previous research has shown that indoor dust contaminated with OPFR is significantly cytotoxic to human corneal epithelial cells (HCECs) [27]. As a high-frequency OPFR found in indoor dust, the toxic effect of TPP on corneal epithelial cells should be evaluated, but few studies have focused on the damage to the eye caused by TPP.

In this study, human normal corneal epithelial cells (HCECs) were used to explore the effects of TPP on the human eye. After exposure to TPP for 24 h, the HCECs were detected for cell viability, morphology, apoptosis, and mitochondrial membrane potential. In addition, the mRNA level of the apoptosis-related marker (*Cyt c*, *Caspase-9*, *Caspase-3*, *Bcl-2*, and *Bax*) was detected to explore the molecular mechanism of TPP that induced HCEC cytotoxicity.

## 2. Results and Discussion

### 2.1. TPP Suppressed Cell Viability and Altered Cell Morphology

Cell viability can effectively reflect the toxic effect of flame retardants on cells [28]. The ocular surface, as the first barrier for human eyeballs, is in direct contact with particles in the air. The human corneal epithelium is the outermost cell on the ocular surface, and is easily damaged by long-term continuous contact with flame retardants in the environment. Xiang et al. (2017) found that HCECs showed a 16% cell viability loss after exposure to 316 μM organophosphorus flame retardants—TDCPP [29]. TPP is one of the OFPRs detected at high rates of concentration in the environment [30]. In this study, our data indicated a dose-dependent decrease in cell viability after TPP exposure (Figure 1). However, at 100–200 μM TPP, the cell viability was decreased from 80% to 56%. When TPP concentration was increased to 400 μM, cell viability sharply decreased, leaving less than 25% of the cells alive. The fitted curve shows that the IC_50_ of TPP at 220 μM (Figure 1) was lower than that of TDCPP at 465 μM. This indicates that HCECs are more sensitive to TPP than TDCPP. Therefore, it is also necessary to explore the mechanism of the toxicity of TPP on HCECs, which will help to identify the harm caused by different organophosphorus flame retardants to the human cornea.

Cell morphology also was used as an important index to determine cell physiological dysfunction and cytotoxicity [29]. The shape of normal corneal epithelial cells is generally pebble-like or a square polygon, and the cells are closely arranged. In this study, the results showed that there was no obvious change in cell morphology until the concentration of TPP >50 μM (Figure 2a–e). This indicates that after low-concentration TPP exposure, the self-defense mechanism of HCECs is activated, making the cells themselves have a certain degree of tolerance to TPP, and basically maintaining the dynamic balance between cells and the outside world [31]. At 100 μM, the number of cells is slightly reduced, the cell arrangement is vacant, the cell morphology is irregular, and the edge appears to have elongated sharp corners, and vacuoles appear (Figure 2f). When exposed to TPP at 200 μM and 400 μM, the number of cells decreased sharply and the outline was rough, the cells overall were elongated in a spindle shape or atrophy in an irregular polygon, and many cells became rounded and floated in the medium (Figure 2g–h). The cell morphology change was coincidental with the cell viability in Figure 1, indicating that the toxicity of TPP to HCECs increased in a dose-dependent manner. To further explore the toxic mechanism of TPP on HCECs, we selected IC_50_ = 200 μM, 1/10 IC_50_ = 20 μM, and 1/100 IC_50_ = 2 μM as the values for the next experiment.

### 2.2. TPP Induced Apoptosis and Changed Mitochondrial Membrane Potential in HCECs

Apoptosis plays an important role in cell death induced by various environment harmful substances [32], and excessive cell death will induce related ocular surface diseases [33]. Previous studies have shown that exposure to indoor dust extraction solution contaminated with organic phosphorus flame retardants caused significant cell apoptosis in HCECs, indicating that cell apoptosis is an important indicator reflecting the cytotoxic effect of organophosphorus flame retardants on HCECs [27]. In this study, Annexin-V FITC/PI double staining was used to stain HCECs. The results showed that the apoptosis rates of HCECs were increased in a concentration-dependent manner after exposure to 2–200μM TPP (Figure 3). HCECs showed a slight change in the cells in the Q2/Q3 quadrant after exposure to 20 μM TPP, and the apoptosis rate increased to 3.84% (Figure 3a,c,e). After treatment of 200 μM TPP for 24 h, the proportion of cells in the Q2/Q3 quadrant was increased significantly, and the apoptosis rate increased to 13.6%, which was 3.5 times higher than that of the control group (2.59%) (Figure 3a,d,e). These results suggest that TPP may induce apoptosis of HCECs, and the degree of damage to HCECs is positively correlated with the concentration, which was consistent with the findings of Xiong et al. (2023), who confirmed that TPP exposure induced EPC cell apoptosis in a dose-dependent manner [34].

Mitochondria are often-targeted cellular organs for the analysis of cellular damage caused by air pollutants [35] and are also deemed as another indicator to reveal the underlying mechanisms of cell apoptosis. They have the basic function of providing energy for cell activity and controlling the process of cell death [36]. The change of mitochondrial transmembrane potential (MTMP) is an important index to evaluate mitochondrial dysfunction in human cells, and a key factor to trigger cell apoptosis. JC-1 is commonly used as a fluorescent probe in MTMP detection. When the mitochondrial membrane potential increases, JC-1 accumulates in the mitochondrial matrix, forming a polymer (j aggregates) that produces red fluorescence. When the mitochondrial membrane potential decreases, JC-1 will disperse in the mitochondrial matrix, and the formed JC-1 monomer produces green fluorescence [37]. In this way, changes in mitochondrial membrane potential can be visually detected by fluorescence color transformation. To prove whether the apoptosis of HCECs induced by TPP exposure is related to mitochondrial damage, we used inverted fluorescence microscopy combined with a JC-1 probe to further study the changes of membrane potential on the mitochondria of HCECs after exposure, to test whether it affects mitochondrial function.

As shown in Figure 4, with the increase in TPP exposure concentration, the red fluorescence intensity decreased (Figure 4e–h), and the green fluorescence signal increased (Figure 4i–l) compared with the control group. At 200 μM, MTMP dissipation was obvious, which was manifested as a decrease in red fluorescence (Figure 4h) and a significant increase in green signal (Figure 4l). Combined with the statistical diagram of green fluorescence (Figure 4q), it can be seen more clearly that with the increase in exposure concentration, the green fluorescence intensity increases in a concentration-dependent manner and significantly increased at 200 μM, which was about 2-fold that of the control group. This was consistent with the cytotoxicity results of Xiang et al. (2018), which showed that HCECs induced a significant reduction in mitochondrial membrane potential through dust exposure solution [38]. In summary, we analyzed whether TPP exposure to HCECs can aggravate MTMP abnormalities, causing decreased viability, changed cell morphological, and induced apoptosis of HCECs, which may be related to changes in mitochondrial membrane potential.

### 2.3. TPP Affects Expression Level of Genes Associated with Apoptosis in HCECs

To further investigate the molecular mechanism underlying the apoptosis induced by TPP and based on the detection results of apoptosis and mitochondrial membrane potential, we hypothesized that the caspase-dependent mitochondrial pathway may be involved in TPP-induced cell apoptosis in HCECs. Cytochrome c (Cyt c), located in the mitochondrial membrane, plays an important role in the intrinsic pathway of apoptosis. The upregulation of Cyt-c indicates that it is released from the mitochondrial inner membrane into the cytosol, and is subsequently involved in the activation of caspase 9/3-induced cell apoptosis [39,40,41]. Bcl-2 family proteins are important factors in the regulation of the mitochondrial apoptosis pathway, and the change of the expression ratio of Bcl-2 and Bax is related to the change of mitochondrial membrane potential. Bcl-2 is an anti-apoptotic factor, and Bax is a pro-apoptotic factor. The increased expression ratio of Bcl-2 and Bax stimulated the release of Cyt c from the mitochondria, thus promoting cell apoptosis [42]. Therefore, qRT-PCR was used to detect the mRNA levels of apoptosis-related genes (*Cyt c*, *Caspase-9*, *Caspase-3*, *Bcl-2*, and *Bax*). Our results showed that the *Bcl-2* mRNA levels decreased by 0.3 to 0.5 times (Figure 5a) and *Bax* mRNA levels increased by 1.6 to 3.3 times (Figure 5b) compared with the control group, respectively. In addition, the mRNA levels of *Cyt c* (2.6-fold), *Caspase-9* (1.7-fold), and *Caspase-3* (2.7-fold) were significantly increased after exposure to 200 μM TPP compared with the control group (Figure 5c–e). These results are consistent with the apoptosis and mitochondrial membrane potential changes of HCECs exposed to TPP. In conclusion, the toxic effect of TPP on HCECs is mainly realized through the mitochondria-mediated apoptosis pathway, which was consistent with the results of previous studies, indicating that TPP treatment induced an obvious cell apoptosis in human normal liver cells (L02) and A549 cells [21,43]. Our results indicate that TPP exposure caused cell apoptosis, and this may be via the caspase-dependent mitochondrial pathway in HCECs, which may provide a new perspective on TPP-induced ocular surface damage.

However, the mechanism of toxic effects of TPP exposure on HCECs is still incomplete. Whether there are other signaling pathways involved is unknown. To further explore the mechanism of toxic effects of TPP on HCECs, the subsequent studies should consider the toxicity effects of endoplasmic reticulum stress and death receptor pathways. Further research on the molecular mechanism of TPP-induced HCECs damage is required.

## 3. Materials and Methods

### 3.1. Chemicals and Reagents

DMEM/F-12 Basic medium and fetal bovine serum FBS are products that we obtained from Wuhan Procell Life Technology Co., Ltd (Wuhan, China). Antibiotic-antifungal solution (100×) and epidermal growth factor (EGF) are products that we obtained from Life Technologies in the United States. An analysis of pure-grade chloroform and isopropyl alcohol came from the Guangdong Chemical reagent Engineering Technology Research and Development Center. Anhydrous ethanol was purchased from Guangdong Guanghua Technology Co., LTD (Shantou, China). The CCK-8 cell viability assay kit, SYBR green qPCR master mix Kit, Cell Cycle assay Kit, and Annexin V-FITC/PI Apoptosis assay Kit were purchased from Nanjing Yi Fei Xue Biotechnology Co., Ltd. (Nanjing, China). Triphenyl phosphate (TPP, purity 99%) was obtained from Dr. Ehrenstorfer GmbH (Augsburg, Germany). Dimethyl sulfoxide (DMSO; >99.9%) was obtained from Sigma-Aldrich (St. Louis, MO, USA). The final concentration of DMSO used in the exposure medium was 0.1% (*v*/*v*) during the experiment.

### 3.2. Cell Viability, Morphology, and Apoptosis

Human Corneal Epithelial Cells (HCECs) were provided by Qin-Xiang Zheng, Deputy Director of Optometry Hospital affiliated with Wenzhou Medical University. HCECs were cultured in DMEM/F-12 medium supplemented with 10% fetal bovine serum, and 10 ng·mL-1 EGF and 1% Antibiotic-Antimycotic in a 5% CO_2_ incubator at 37 °C and were passaged every 2 days. The cells were digested and seeded into the 96-well plate at a density of 1 × 10^4^ cells per well and cultured at 37 °C for 24 h. Cultured cells were placed in different concentrations (6.25, 12.5, 25, 50, 100, 200, and 400 μM) of TPP over 24 h, and, following their exposure, we used an inverted microscope (TS100, Nikon, Tokyo, Japan) to observe cell morphology and pictures were taken to record our findings. Then, 10%CCK8 solution was added and incubated in a constant temperature incubator (5% CO_2_, 37 °C) for 2 h. The absorption values were measured at 450 nm using SpectraMax^®^Plus 384, Molecular Devices, San Jose, CA, USA, and the percentage values of cell viability were calculated. The nonlinear simulation between TPP concentration and cell viability was performed using GraphPad Prism 8 software. The half maximal inhibitory concentration (IC50) of exposure to corneal epithelial cells was calculated.

HCECs on 6-well plates were exposed to TPP at 37 °C for 24 h at concentrations of IC_50_ = 200 μM, 1/10 IC_50_ = 20 μM, and 1/100 IC_50_ = 2 μM, respectively. After washing with PBS, the cells were digested with trypsin. The centrifugally cleaned cells were added to a 100 μL binding buffer, with Annexin V-FITC 5 μL and propyl iodide 2.5 μL, and were then incubated at room temperature for 15 min away from light. Finally, the apoptosis was detected using a flow cytometry (CyFlow^®^Cube 6, Patec, Nuremberg, Germany). The results were analyzed by FlowJo_V10 software.

### 3.3. Mitochondrial Membrane Potential Assay

HCECs were inoculated in 6-well plates and exposed at 0, 2, 20, and 200 μM TPP for 24 h. Subsequently, the cells were washed once with PBS, added to 1 mL of DMEM and 1 mL of JC-1 staining working solution, thoroughly mixed, and incubated in the cell incubator at 37 °C for 20 min, during which the configured JC-1 staining buffer (1×) ice bath was added. After incubation, the supernatant was sucked, washed twice with the JC-1 staining buffer (1×), then 2 mL of DMEM was added, and the pictures were observed under the fluorescence microscope and recorded.

### 3.4. Cell Total RNA Extraction and cDNA Synthesis and Quantitative Real-Time PCR Analysis

HCECs were inoculated in 6-well plates and exposed at 0, 2, 20, and 200 μM TPP for 24 h. Further RNA was isolated by Trizol reagent according to the manufacturer’s instructions. The concentration and purity of purified RNA were measured by the Nano Photometer^®^ N60 (IMPLEN GmbH, Munich, Germany). About 1 μg total RNA was reverse transcribed to cDNA via the Script 1st Strand cDNA Synthesis Kit (Yi Fei Xue Biotechnology Co., Ltd., Nanjing, China), and the reverse transcriptional conditions were as follows: 50 °C for 15 min, 75 °C for 5 min, and finally stopped at 4 °C. Later, real-time quantitative PCR was performed according to the requirements of the SYBR Green qPCR main mixing kit. Reaction conditions were as follows: (1) Template predenaturation: 95 °C, 2 min. (2) Amplification: 95 °C, 10 s, 60 °C, 30 s, 40 cycles; 95 °C, 5 s. (3) Welding curve: 65 °C, 5 s, 95 °C, 0.5 °C/s. Then, β-Actin was used as an internal reference gene, and the expression of related genes was calculated by the 2^−ΔΔCt^ method. The specific primers are shown in Table 1.

### 3.5. Statistical Analysis

All experiments were independently repeated three times, and the results were ex-pressed as mean ± standard deviation (M ± SD). Data were processed by Graphpad Prism 8.0 software, and one-way ANOVA was performed on the results. Significant difference was defined as ** *p* ≤ 0.05. FlowJo_V10 software was used to plot the apoptosis rate. Im-age-Pro Plus 6.0 software was used to process fluorescent images.

## 4. Conclusions

In this study, we confirmed that TPP induced toxicity in HCECs, and its concentration is positively correlated. We found that TPP induced cell apoptosis of HCECs and reduced the mitochondrial membrane potential in a concentration-dependent manner. In addition, TPP increased the expression of *Cyt c*, *Caspase-9*, *Caspa-se-3*, and *Bax* and suppressed the expression of *Bcl-2* in HCECs, which was validated by a RT-qPCR analysis. In summary, our results show that TPP suppresses cell viability through inducing apoptosis in a mitochondria-dependent manner and that this induction may be mediated by activating caspase activity, upregulating *Bax*, and downregulating *Bcl-2*. The research results can provide theoretical support for evaluating the damage of organophosphorus flame retardants, namely TPP, to human corneal health. In addition, such results can also provide a reference for an overall assessment of the damage to corneal health caused by organophosphorus flame retardants and provide scientific evidence for effectively preventing the harm of organophosphorus flame retardants caused to human corneas.

## Figures and Tables

**Figure 1 ijms-25-04155-f001:**
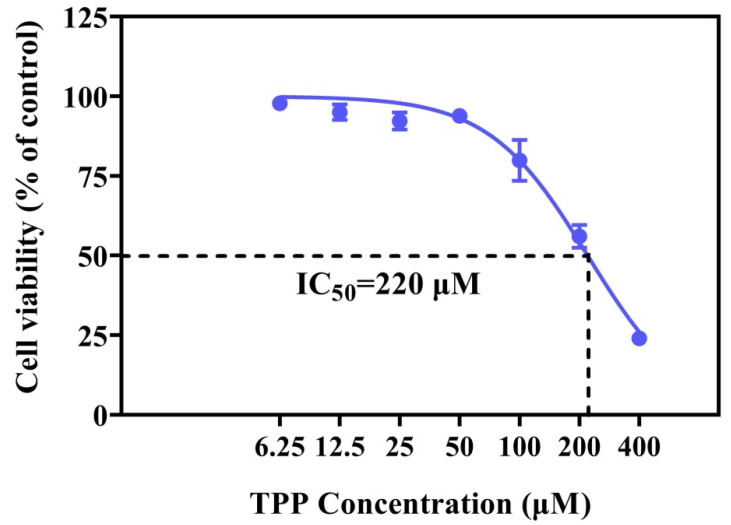
The changes in cell viability after TPP exposure for 24 h in HCE cells. IC_50_ was determined by nonlinear regression analysis.

**Figure 2 ijms-25-04155-f002:**
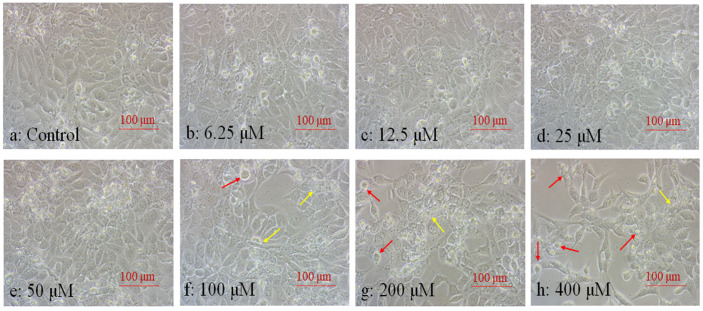
Morphological changes of HCECs after 24 h treatment of 6.25–400 μM TPP. The inverted microscope is lowered to a large 100× recording image. Red arrows point to vacuoles and floating cells, and yellow arrows point to atrophic cells.

**Figure 3 ijms-25-04155-f003:**
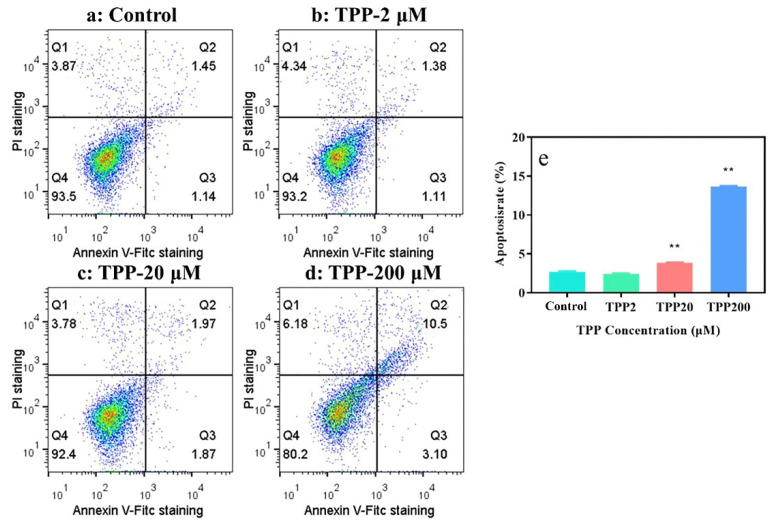
TPP-induced HCEC apoptosis based on flow cytometry: cell apoptosis rate (**a**–**d**) after exposure to 0–200 μM TPP for 24 h. The bar chart is the sum of the Q2 and Q3 quadrants (**e**), each of which is the average of three replicates ± SEM. ** *p* < 0.01 compared with the control group.

**Figure 4 ijms-25-04155-f004:**
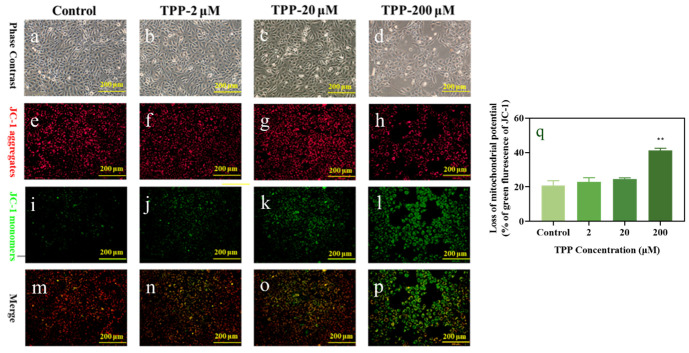
TPP-induced changes of mitochondrial transmembrane potential (MTMP) in HCECs. After the cells were exposed to 2–200 µM TPP for 24 h, the cells were stained with a JC-1 probe and images were acquired under an inverted fluorescence microscope with 200× magnification. The images include the cell morphology after exposure (**a**–**d**), the red fluorescence of the JC-1 polymer (**e**–**h**), the green fluorescence of the JC-1 monomer (**i**–**l**), the synthesis of the two colors of fluorescence (**m**–**p**), and the statistical change of the green fluorescence content (**q**). ** *p* < 0.01 compared with the control group.

**Figure 5 ijms-25-04155-f005:**
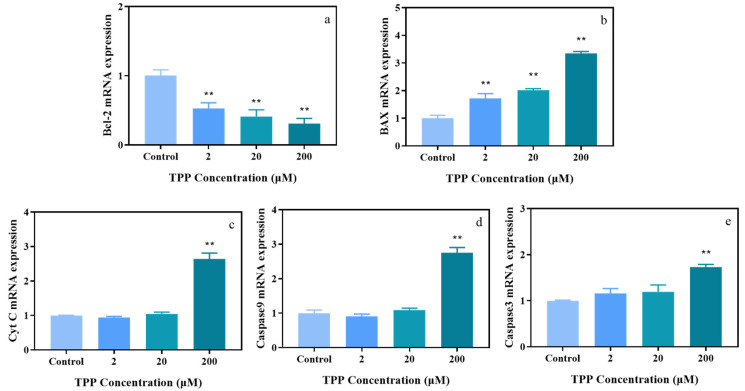
TPP alters the expression of apoptosis-related genes after exposure to 2–20 μM TPP for 24 h in HCECs. The relative expression levels of *Bcl-2* (**a**)*, Bax* (**b**), *Cyt c* (**c**), *Caspase 9* (**d**), and *Caspase 3* (**e**) were detected by RT-qPCR. ** *p* < 0.01 compared with the control group.

**Table 1 ijms-25-04155-t001:** The primers for RT-qPCR.

Primer Name	Forward Primer (5′-3′)	Reverse Primer (5′-3′)
β-actin	GACATCCGCAAAGACCTG	GGAAGGTGGACAGCGAG
Bcl-2	CATGGAAGCGAATCAATGGACT	CTGTACCAGACCGAGATGTCA
Bax	AAACTGGTGCTCAAGGCCC	CTTGGATCCAGACAAGCAGC
Caspase-3	CATGGAAGCGAATCAATGGACT	CTGTACCAGACCGAGATGTCA
Caspase-9	CTCAGACCAGAGATTCGCAAAC	GCATTTCCCCTCAAACTCTCAA
Cyt c	AGGCCCCTGGTACTCTTACACA	TCTGCCCTTTCTTCCTTCTTCTTA

## Data Availability

Data are contained within the article.

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
