# Peer review of "Organophosphorus Flame Retardant TPP-Induced Human Corneal Epithelial Cell Apoptosis through Caspase-Dependent Mitochondrial Pathway"

_ijms, 2024, doi:10.3390/ijms25084155_

Round 1

Reviewer 1 Report

Comments and Suggestions for Authors

Title: Organophosphorus flame retardant TPP-induced human corneal epithelial cell apoptosis through mitochondrial apoptosis pathway

With the respect of the manuscript I have few concerns with the manuscript which does not affect the core plot of the manuscript. Hence, I recommend the manuscript for publication after major revision.

The entire manuscript has lot of jargon complex through sentence, check the grammatical error

Introduction

1-      Author must rewrite intro part due to the lack rational study objective.

2-      In line number 55 What L02 ?   

Materials and Methods

3-      Line number 214 author should be changed IC50

4-      3.4 author provide the details methodology for Cell total RNA extraction and cDNA synthesis and Quantitative real time PCR analysis.

5-      Results

6-      In cell viability and morphology author should be details description of cell morphology and what did you observe the changes are must be clearly stated in the revision

7-       Line number 106 to 120 Apoptosis results only discussion, you have good result why did you wrote result in  single line; author must attention  to rewrite the brief description on your results control and other concentrations in Figure-3 compared with each quadrate. 

8-      Author attention to need the detailed discussion with compare with your results not needed to previous study.

9-      Author, should be clearly mention the scale bars in Figure-4 a and check carefully recheck the statistical error analysis.

10-  Author should be changed the Figure-3 due to the poor quality of images.

11-  Figure -2 is very poor quality author must change the Figure-2

12-  In Fig-2 control image and treated 50µM are same morphology author attention to change the image

13-  Why all cell line images have red color line in the right bottom, I think its look like scale bar. Author should be explanation.

14-   In Figure-5 author must check the statistical error bar

15-  Author provide the gel image for gene expression Caspase-9, Caspase-3, Bcl-2, and BAX

16-  Which software using in statistical analysis

Conclusion

17-  Conclusion look like result, author should be rewrite clear conclusion of your study objective.

Reference

18-   All reference add a DOI number 

Comments on the Quality of English Language

Minor English language editing required

Reviewer 2 Report

Comments and Suggestions for Authors

Chen et al. introduce an article presenting original results of a research study on Organophosphorus flame retardant TPP-induced human corneal epithelial cell apoptosis through the mitochondrial apoptosis pathway. The authors are appreciated for planning and performing this research study and then presenting it, including writing the introduction, materials and methods, data generation, results, statistical analysis, and conclusions. However, the presentation needs significant improvements to make it more engaging for readers. Some suggestions for improvement include:

  • -The permissible concentration of TPP, non-toxic for health, as recommended by health agencies, is not mentioned anywhere in the manuscript.

  • -Clarify the meaning of "(such as conjunctivitis and cornea)".

  • -In Section 2.1. Cell viability and morphology, Lines 72-79 appear to be part of the introduction or discussion, not the results.

  • -Figure 1A and B present the same results. It is recommended to delete Figure 1A.

  • -Arrows should be included in Figure 2f and 2g-h to indicate vacuoles, atrophy, and floating cells.

  • -In Section 2.2. Apoptosis and mitochondrial membrane potential, the first paragraph should be adjusted somewhere else, starting directly with the description of the results.

  • -The next paragraph (Lines 128-140) needs to be adjusted somewhere else in the manuscript. Similar text can be found in other parts of the results. It seems that results and discussion are written together in Section "2. Results".

  • -The limitations of the study, future work, and alternatives or opinions regarding the safety of TTP should be well described. How these research results could be applicable in a clinical setup should also be discussed to capture the attention of readers.

  • -It is not very clear how this research work is different from other published current research works using different chemicals. Why should one focus only on this? Why not on others which belong to organophosphorus flame retardants?

Overall, the paper needs a thorough revision to improve the presentation of the research work.

Reviewer 3 Report

Comments and Suggestions for Authors

The manuscript is well written, and the results have implications for human health, as concluded by the authors. A careful review was performed, and few issues were found. Please see the comments below. I believe that addressing these comments will further improve the quality of the manuscript.

1.…………”to explore the mechanism of toxic of TPP on HCEC………..

2. Our found that there were no significant changes in……………………

3. In Figure 3b. Please bring the caption to the axis closure.

4. The authors conclude that the cells in Figure 2 reflect the stated changes after the treatments. Please include a magnified image to visualize the changes mentioned in the result. Here, staining with specific antibodies will be helpful. For example, Actin/Phalloidin and Dapi. Quantification may be helpful.

5. Please correct the sentence …The apoptosis results that there was no significant in apoptosis rate compare………..

6. Please check if MTMP in place of  MMP would be suitable to denote themitochondrial transmembrane potential.

7. Please reorganize this sentence as this is difficult to understand…..”However, TPP no changed 177 apoptosis-related genes mRNA levels at the low-concentration, which may be low concentration TPP did not………………………………

8. Please check this sentence “This is consistent with previous studies on mitochondrial pathway apoptosis in PHNCs.

9. Please check this sentence “Therefore, it is concluded that the toxic of TPP induced HCE cells is mainly through the mitochondrial pathway”.

10. The authors may change the subheading “ . Mitochondrial apoptosis-related genes analysis” as it appears that they are checking the apoptosis of mitochondria but not the HCS cells. However, as per the experimental design, they are analyzing the apoptosis of tis cell, which is regulated by mitochondrial genes. Please confirm.

11. The title is confusing as there are no mitochondrial apoptosis shown. In fact, the study shows that HCS cell apoptosis is caused by changes in Mitochondria ( membrane potential or gene level). Please confirm. Moreover, the RNA for the RT PCR was derived from the cellular lysates not from mitochondria. Please confirm.

12. Moreover, There are limited experiments to show mitochondrial dysfunction. A Change of mitochondrial transmembrane potential (MMP) is an important index to evaluate mitochondrial dysfunction. However, at least one additional experiment focusing on mitochondrial function should complement this conclusion.

13. Analysis of mitochondrial-specific genes would be beneficial. Also, the expression level of apoptotic protein-cleaved caspase 3 and total caspase 3 would be helpful in showing apoptosis. 

Comments on the Quality of English Language

A careful revision of the writing is needed. There are issues in several places. Please check the writing carefully.

Round 2

Reviewer 1 Report

Comments and Suggestions for Authors

Authors revised the manuscript based on the reviewer comments and so the manuscript maybe accepted in its present form. 

Reviewer 2 Report

Comments and Suggestions for Authors

The revised manuscript is significantly improved from the previous version, and it can now be considered to attract readers to engage with the scientific content of the paper. Although there are some grammatical errors in the text, particularly in the last few paragraphs, they can be corrected through proofreading. Therefore, this current paper is accepted for publication.

Reviewer 3 Report

Comments and Suggestions for Authors

Thank you for addressing  the comments and providing suitable justifications for some of the comments. After careful review of the updated manuscript it showed that there are some minor issues as listed below.

1. The authors wrote that they have now used MMT as an abbreviation for mitochondrial transmembrane potential in the manuscript and cited line no 207 in their response, but it is not found to be addressed. Please carefully check and address this issue. The author should use the abbreviation consistently. MMT is not suitable as well. MTMP has been used previously and hence can be used for this study.

2. There are some errors in sentences. For example, the sentence in line 310 is incomplete. Please read the entire manuscript thoroughly to correct any mistakes.

3. In the method section, line 409, Please correct the sentence "About 106 HCECs were inoculated in 6-well plates, and exposure 0, 2, 20, 200 μM 409 TPP for 24 h. Then total RNA was separated by Trizol reagent according to manufactur-..... Here it could be written as HCECs were inoculated in 6 well plates and exposed at 0, 2....... 

Further RNA was isolated ..........

Comments on the Quality of English Language

Dear Editor,

Thank You. There are some minor issues in the writing. I recommend a minor revision to address them. The authors should carefully check the entire manuscript.
